# Dynamic Characteristics Monitoring of Large Wind Turbine Blades Based on Target-Free DSST Vision Algorithm and UAV

Wanrun Li [1,2,3,]*[ID], Wenhai Zhao [1][ID], Jiaze Gu [4][ID], Boyuan Fan [1][ID] and Yongfeng Du [1,2,3]

1   Institution of Earthquake Protection and Disaster Mitigation, Lanzhou University of Technology, Lanzhou 733050, China; zhaowh@lut.edu.cn (W.Z.); fanby@lut.edu.cn (B.F.); dooyf@lut.edu.cn (Y.D.)
2   International Research Base on Seismic Mitigation and Isolation of GANSU Province, Lanzhou University of Technology, Lanzhou 733050, China
3   Disaster Prevention and Mitigation Engineering Research Center of Western Civil Engineering, Lanzhou University of Technology, Lanzhou 733050, China
4   School of Computer and Communication, Lanzhou University of Technology, Lanzhou 733050, China; gujiaze@lut.edu.cn
*   Correspondence: ce_wrli@lut.edu.cn

**Abstract:** The structural condition of blades is mainly evaluated using manual inspection methods. However, these methods are time-consuming, labor-intensive, and costly, and the detection results significantly depend on the experience of inspectors, often resulting in lower precision. Focusing on the dynamic characteristics (i.e., natural frequencies) of large wind turbine blades, this study proposes a monitoring method based on the target-free DSST (Discriminative Scale Space Tracker) vision algorithm and UAV. First, the displacement drift of UAV during hovering is studied. Accordingly, a displacement compensation method based on high-pass filtering is proposed herein, and the scale factor is adaptive. Then, the machine learning is employed to map the position and scale filters of the DSST algorithm to highlight the features of the target image. Subsequently, a target-free DSST vision algorithm is proposed, in which illumination changes and complex backgrounds are considered. Additionally, the algorithm is verified using traditional computer vision algorithms. Finally, the UAV and the target-free DSST vision algorithm are used to extract the dynamic characteristic of the wind turbine blades under shutdown. Results show that the proposed method can accurately identify the dynamic characteristics of the wind turbine blade. This study can serve as a reference for assessment of the condition of wind turbine blades.

**Keywords:** structural health monitoring; large wind turbine blades; UAV; target-free; DSST vision algorithm

## 1. Introduction

With the world's increasing consumption and utilization of energy, the development and utilization of renewable energy has gradually gained attention, to lessen the adverse global impacts of non-renewable energy. Among the many renewable energy sources that meet the current sustainable development and future international strategies, wind energy has been spreading because of its advantages of clean use, small cost and low impact on the environment [1]. As one of the main forms of energy conversion from wind into electric, wind power generation is deeply appreciated by people. As an important component of wind turbine structure power conversion, the state of the blade directly affects the performance and power quality of a wind turbine. Once the wind turbine blade is damaged, it will not only cause accidents but also serious economic losses [2–4]. Therefore, accurate monitoring of blades for health assessment is of great significance to the operation, maintenance and safety of wind turbine structures.

Traditional wind turbine blade monitoring uses high-power telescope detection, acoustic emission monitoring [5], ultrasonic monitoring [6], thermal imaging monitoring [7], etc. However, all the above methods are limited by human experience, equipment accuracy

and environmental impact so the effect is not ideal in actual blade monitoring. In the field of civil engineering, contact sensors are usually used in monitoring structures, but installing that sensors on structures, especially the larger ones, requires substantial logistics support, such as arranging lines in advance, planning sensor locations, installing sensors, maintaining data transmission and other technicalities, which undoubtedly increase the monitoring cost. In addition, the installation of contact sensors on the structure will undoubtedly affect the dynamic characteristics of the object structure. Besides that, the monitoring data depends strictly on the number of sensors [8]. Therefore, it is not suitable for large-scale structural health monitoring under numerous wiring arrangements. With the development of science and technology, wireless sensors have emerged in structural health monitoring [9]. However, due to their disadvantages such as high cost, time-consuming installation, and asynchronous data reception, they cannot be widely used in wind turbine blade monitoring. Most wind farms are located in remote areas [2], such as coastal areas, mountain regions, deserts, etc. while the blades are usually located at a high distance from the ground. Under severe natural conditions such as thunderstorms, storms, and salt fog, the traditional monitoring equipment would have low survival rates and high maintenance cost. Therefore, there is an urgent need to develop a monitoring method for a large-scale wind turbine blade with low cost, convenient installation and high reliability.

In recent years, with the rapid development of computer technology and image processing technology, computer vision technology stands out in structural health monitoring with its advantages of low cost, non-contact and non-damage properties. Many scholars have applied computer vision technology to the structural health monitoring of civil engineering [10–12]. Yang et al. [13] proposed a video-based method to identify the micro-full-field deformation mode and the dominant rigid body motions, but it was affected by illumination and obstacles. Dong et al. [14] proposed a method for measuring velocity and displacement based on feature fixed matching and tested it on a footbridge, but its accuracy was greatly affected by illumination. Zhao et al. [15] developed a color-based bridge displacement monitoring APP, which was sensitive to light and did not consider complex background, so that it was not conducive to long-term monitoring. All the above monitoring methods are still unable to accurately monitor the light changes and accommodate complex backgrounds, which are very common in the actual situations of wind turbine blades monitoring. Therefore, the above vision algorithm cannot settle the problem discussed before.

Although computer vision technology is booming in civil engineering structural health monitoring, most studies only use one or several cameras with fixed positions for monitoring [16,17]. The obtained data using a fixed camera depend heavily on image quality, which makes it impossible to take a comprehensive picture of large structures. The image distortion caused by the zoom due to atmospheric effect, and the inability to determine the most favorable position of the required monitoring points makes it difficult to monitor the dynamic characteristics of large structures by use of cameras [18]. Because of the flexibility and convenience of UAV, it has become popular in the field of vision-based structural health monitoring. More and more scholars do research on vision-based structural health monitoring by UAV. Bai et al. [19] adopted the template matching computer vision method of normalized cross correlation (NCC) to monitor structural displacement, and used the background points to obtain the vibration of UAV for displacement compensation. Zhao et al. [20] proposed a dam health monitoring model based on UAV measurement, which artificially creates damage to the dam and uses the measurement point cloud to measure the damage. Perry et al. [21] proposed a new remote sensing technology for dynamic displacement of three-dimensional structures, which integrates optical and infrared UAV platforms to measure the dynamic structural response. Wu et al. [22] extracted blurred images from videos of wind turbine blades under rotation, and used adversarial generative networks to expand the blade data for the purpose of defect detection, but dynamic characteristics were not tested. Targets are used in the above methods to enhance the features so that the structural displacement will be identified.

Although many scholars use target-free methods to monitor structural health [23,24], they still find the inherent patterns or nuts on the surface of the structure for identification. There are no obvious textures and patterns on the surface of large-scale wind turbine blades, which are different from the overall structure. In addition, it is impossible to attach targets to the blades. Therefore, a non-contact and target-free monitoring method must be sought for the visual monitoring of large-scale blades.

This paper is focus on monitoring the dynamic characteristics of large wind turbine blades based on vision algorithms and UAV. In the first section, the disadvantages of traditional monitoring methods and the limitations of existing vision monitoring methods in the monitoring of large wind turbine blades are summarized. Furthermore, a method for structural health monitoring of large wind turbine blades based on target-free DSST vision algorithm and UAV is proposed. The second section studies the spatial displacement drift caused by UAV hovering monitoring. The method of the background fixed point combined with high-pass filtering is proposed to eliminate the in-plane influence during hovering monitoring, the adaptive scale factor of which eliminates the UAV out-of-plane displacement drift. The spatial displacement drift of the UAV hovering monitoring is compensated and the corresponding experimental verification is done. In Section 3, a machine learning method is adopted to train the position filter and scale filter of a DSST algorithm, and a target-free DSST visual monitoring method is proposed. In Section 4, experiments are carried out in combination with the actual engineering background illumination transformation and complex background conditions to verify the robustness of the target-free DSST algorithm. In Section 5, the wind turbine blades vibration videos combined with the blades corners and edges tracking are used to identify the dynamic displacement of the blades without additional targets, and the dynamic response of each monitoring point of the blades is obtained through the vision, which is evaluated and compared with the analysis results of traditional monitoring methods. In Section 6, the main conclusions of this work are summarized.

## 2. UAV Space Displacement Compensation

### 2.1. UAV Lens Noise Test

The camera is popular in the vision monitoring field because of its stable performance, long standby time and high resolution. However, for large structures, especially high-rise structures, the camera cannot be used for accurate monitoring due to its large shooting angle and fixed position, making it hard to reach the optimal monitoring position. In view of the disadvantages of fixed camera monitoring, a flexible and convenient UAV is used for structural health monitoring, which can not only monitor at the optimal position, but also maximize the use of image resolution to ensure monitoring accuracy.

Although UAV monitoring has advantages over cameras, its vibration cannot be ignored during the monitoring process. Therefore, the motion drift must be compensated for UAV monitoring during the monitoring process. The UAV carries out vision monitoring by carrying a lens. The test here is conducted to explore the lens noise when the UAV is hovering and monitoring, adopting the resolution of 720P/1080p/4K and the frame rate of 50/60 fps for 60 s. The test equipment is shown in Figure 1.

To accurately measure the hovering monitoring law of the UAV, a stationary marker was set on the wall. Then the DJI Phantom 4Pro UAV was adopted for hovering monitoring compared with the data obtained by the mobile phone and the camera within 60 s. The time history of one point monitoring displacement is shown in Figure 2.

It can be seen from Figure 2 that the placement of the camera and the mobile phone remain basically unchanged during the test time, because this equipment is fixed by a tripod. However, the UAV performs hovering while monitoring during the test. Due to the external environment, such as wind, position and other factors, the UAV moves in space, resulting in displacement drift during monitoring process with a maximum drift of 134 pixels. Hence the lens drift must be adjusted for UAV hover monitoring by converting

the test displacement time history to frequency domain using the fast Fourier transform as shown in Figure 3.

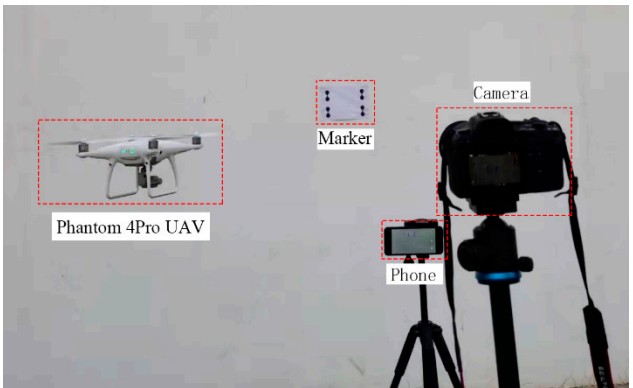

**Figure 1.** UAV hovering monitoring lens noise test equipment.

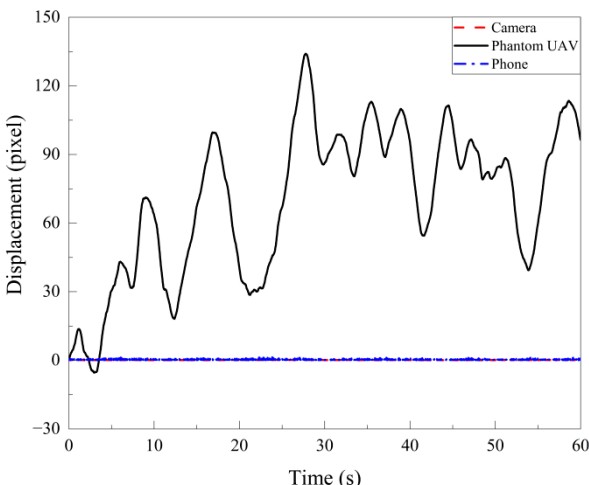

**Figure 2.** Comparison of lens drift time history from different equipment.

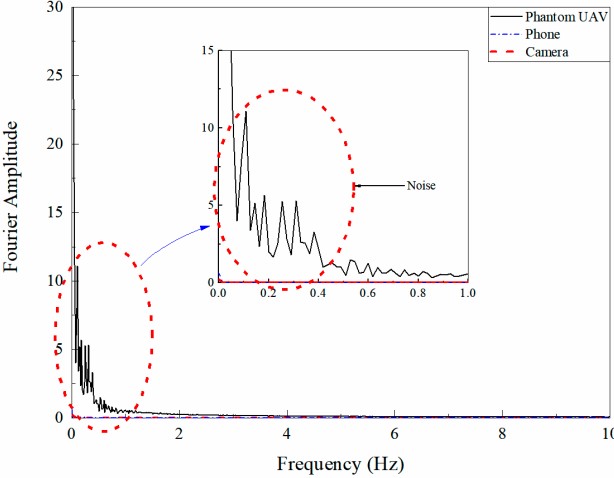

**Figure 3.** Comparison of the lens noises in the frequency domain obtained from different equipment.

It can be seen from Figure 3 that the displacement of the camera and the mobile phone is basically 0, which means there is no noise in the frequency domain, whilst the noise generated by the UAV lens is mostly concentrated in 0~0.5 Hz. Given that the displacement drift is generated in three-dimensional space when the UAV is hovering and monitoring,

the displacement in the X-Y plane and Z plane are adopted in this paper to compensate for the drift generated by the UAV lens.

### 2.2. Displacement Compensation in UAV Hovering Plane

It is now assumed that the UAV only has displacement drift when hovering in the X-Y plane under physical coordinates. Setting a background fixed point in the monitoring image can reflect the movement in the plane when the UAV is hovering. When the UAV moves in the X-Y plane, its movement and the monitoring point during the monitoring process are shown in Figure 4.

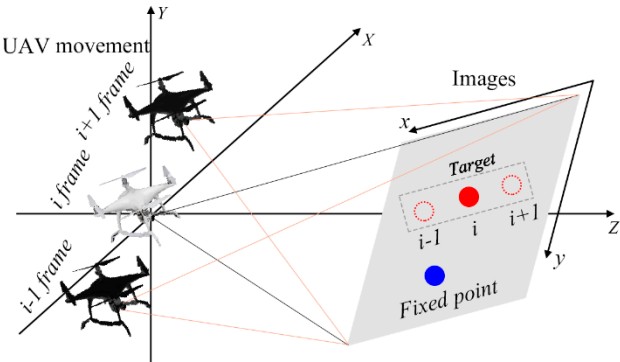

**Figure 4.** Schematic diagram of in-plane displacement drift compensation when the UAV is hovering.

As shown in Figure 4, it is assumed that there is a fixed point that can track a frame of coordinates under hover monitoring and the UAV only moves in the X-Y plane when hovering. The moving point is always in a state of change, so the displacement drift generated by the hovering of the UAV is compensated by the position information of the fixed points. The fixed point in the image captured by the UAV compensates for the hovering motion of the UAV to obtain the absolute coordinates of the monitoring points: the absolute displacement (unit: pixel) of the monitoring structure target when UAV is used can be calculated by Equation (1).

$$P_{AOi} = P_{Oi} - P_{UAVi} = (x_{Oi} - x_{UAVi}, y_{Oi} - y_{UAVi}) \tag{1}$$

where $P_{AOi}$ is the absolute position of the target monitoring point at frame $i$, $P_{Oi}$ is the relative position of the target monitoring point in the frame $i$, and its coordinates are $(x_{Oi}, y_{Oi})$, and $P_{UAVi}$ is a hypothetical fixed point whose coordinates are $(x_{UAVi}, y_{UAVi})$.

However, the actual monitoring conditions are complex. If the fixed-point drifts or the fixed-point is too far away from the monitoring point during the monitoring process, the use of the background fixed point to process the hovering displacement drift of the UAV X-Y plane will cause error and be unable to achieve the purpose of accurate monitoring. Considering the above problems, this paper proposes a method with the combination of background fixed points and high-pass filtering to deal with the displacement drift in the X-Y plane of the UAV; that is, when there are relatively suitable fixed points in actual monitoring, the above background fixed points are used for displacement compensation; otherwise, the relative displacement obtained by monitoring is processed by high-pass filtering, and the main parameters are obtained by using the hovering drift law of the UAV. Finally, the displacement compensation in the X-Y plane of the UAV is achieved.

Figure 5 illustrates the displacement time history which is monitored by vision and verified by LDS when the wind turbine is under the effect of excitation and vibrates freely.

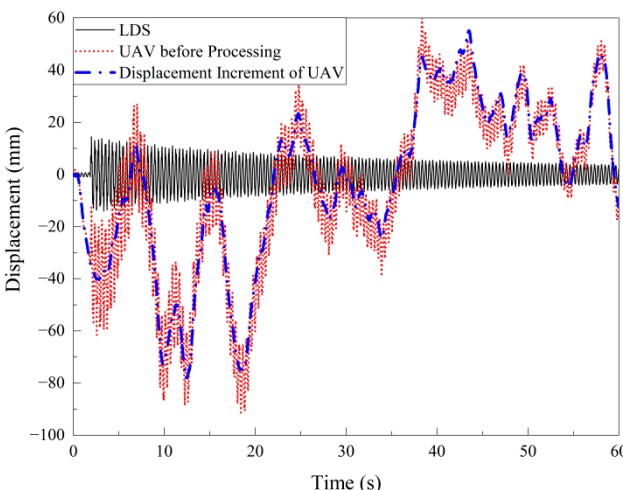

**Figure 5.** Comparison between UAV monitoring structure, background fixed point and LDS monitoring.

It can be seen that if the displacement of UAV hovering monitoring is not processed, the structural monitoring data will be scattered, making the structural response analysis difficult. It is worth noting that the displacement increment of the fixed point is generally consistent with the overall displacement response of the structure, which proves the feasibility of compensating for the displacement drift generated by the hovering of the UAV through the fixed point.

The common way to remove the hovering effect of UAVs is to use high-pass filtering to remove noise. Given that the noise of the UAV's lens measured in Section 2.1 is mainly focused at 0.5 Hz, a high-pass seventh-order ellipse with a cutoff frequency of 0.5 Hz with a roll frequency of 90 dB is used as the filter. The displacement time history obtained by the UAV in the test is denoised with the high pass filter, and the results are shown in Figure 6.

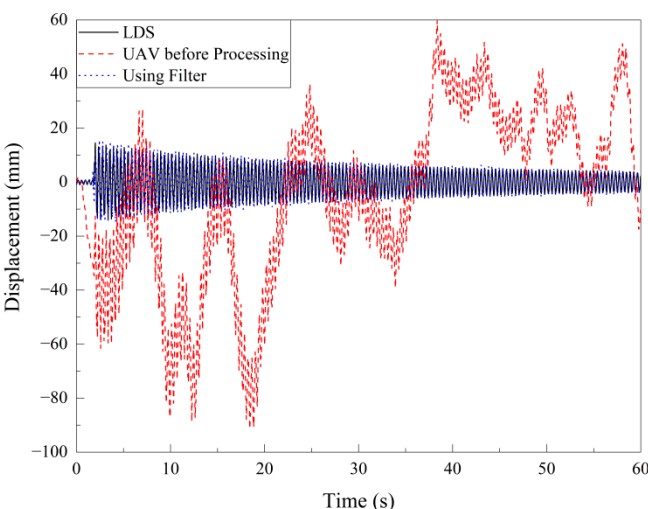

**Figure 6.** Comparison among the original UAV monitoring results, the processing UAV monitoring results and LDS monitoring results.

It can be seen from Figure 6 that the displacement time history curve processed by the high-pass filter is mostly consistent with the LDS monitoring effect, which proves the feasibility of using the high-pass filter to deal with the influence of displacement drift in the plane when the UAV is hovering.

### 2.3. Out-of-Plane Displacement Compensation for UAV Hovering

As known, the UAV moves not only in the X-Y plane, but also the Z-axis outside the plane, so the movement of the Z-axis cannot be ignored, which is also the differ-

ence between monitoring by UAV and camera. Yoon et al. [25] adopted a dynamic scale factor to eliminate the problem of UAV Z-direction motion, and then transformed it into physical coordinates through image coordinates, proposing an adaptive scale factor method based on the assumption that the relative position of the rigid body is invariant. Through the adaptive scale factor, the displacement of structural monitoring is converted from image coordinates to physical coordinates so as to perform out-of-plane displacement compensation.

Figure 7 shows the correspondence between the images captured by the UAV at different times when the UAV only moves in the direction of the Z-axis outside the plane. It is assumed that there are two points in the structure, and the distance between them does not change under the assumption of the relative position of stiffness remaining unchanged. When the UAV moves in the negative direction of the Z-axis, the resolution remains unchanged with the increase of its sight distance so that the original two points with constant distance shrink frame by frame in the image. From Figure 7, it can be observed that the pixels corresponding to the two points gradually decrease when the UAV moves in the negative direction of the Z-axis under the circumstance that the image resolution remains unchanged. In the usual vision-based structural health monitoring, a fixed camera is used for shooting. When image coordinates are converted into pixel coordinates, the scale factor can be set to a certain value as a result of the ignorance of the camera motion [20]. However, when the UAV moves along with the Z-axis, it is no longer possible to use a single scale factor for conversion, and an adaptive scale factor is introduced to compensate for the displacement error caused by the UAV Z-axis movement.

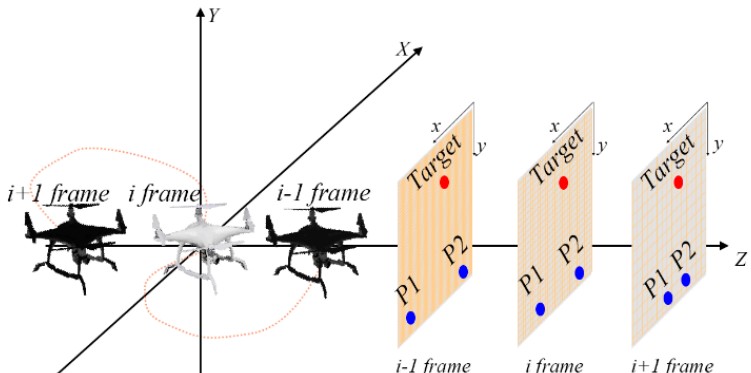

**Figure 7.** Schematic diagram of UAV out-of-plane motion.

Assuming that the actual distance between two points $P_1$ and $P_2$ is $L$ (two points are optional on the structure), the distance between the corresponding two points in the image captured by the UAV is $l$ between $p_1$ and $p_2$. Under the assumption that the relative position of the rigid body remains unchanged, $L$ is a constant value, and $l$ varies with the movement of the UAV in the Z direction. The calculation equation of $l$ is as follows:

$$l = \|p_1 - p_2\| \tag{2}$$

The scale factor of the fixed camera can be determined by the ratio of the actual length of the two points to the pixel length. Due to the Z-direction movement of the UAV, the positions of $p_1$ and $p_2$ change with time. So the adaptive scale factor in each frame is:

$$S_i = \frac{L}{l} = \frac{L}{\|p_{i1} - p_{i2}\|} \tag{3}$$

where $S_i$ is the scale factor at frame $i$, $p_{i1}$ and $p_{i2}$ are the image coordinates of $p_1$ and $p_2$ at frame $i$ respectively.

Combining Equations (1)–(3), the absolute displacement in image coordinates can be transformed into the absolute displacement in physical coordinates:

$$d_{Ai} = S_i(d_{Ri} - d_{UAVi}) \tag{4}$$

where $d_{Ai}$ is the absolute displacement of frame $i$ in house coordinates, $d_{Ri}$ and $d_{UAVi}$ are the relative displacement of frame $i$ in image coordinates and the drift displacement generated by the hovering motion of the UAV, respectively. In the image, the fixed points needed to compensate for the drift of UAV hovering motion can be shared with the two relative fixed points of the adaptive scale factor test. Therefore, to calculate the absolute displacement under physical coordinates, only two fixed points need to be determined.

The UAV, mobile phone, and camera are shot at the same time for two fixed points for 60 s. The ratio of the physical distance to the pixel distance is taken as the scale factor of each frame, which is finally normalized as shown in Figure 8.

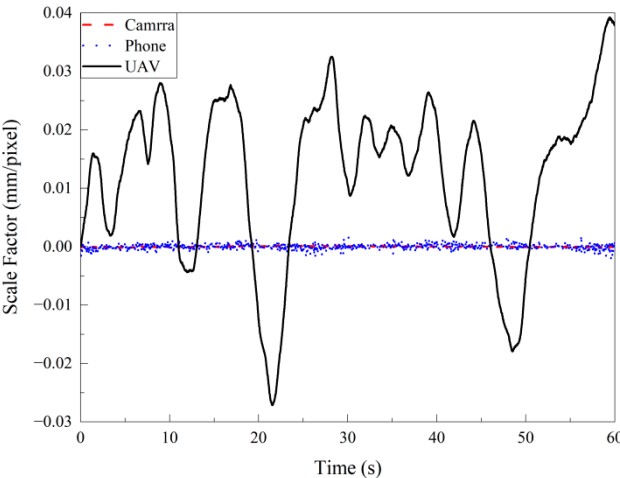

**Figure 8.** Comparison of adaptive scale factors of different devices.

It can be seen from Figure 8 that the adaptive scale factors of the camera and mobile phone float up and down at 0 after normalization, with the floating range not exceeding 0.1%. Therefore, for vision monitoring using fixed equipment, the scale factor of the first frame can be selected for the conversion of image coordinates and physical coordinates. It can be seen from the figure that due to the Z-direction movement of the UAV, the adaptive scale factor fluctuates greatly per frame, and the maximum error can reach 3.93%. Therefore, the monitoring can be more accurate by applying the UAV for visual monitoring and the adaptive scale factor.

The out-of-plane displacement drift of the UAV can be compensated by an adaptive scale factor, and the in-plane displacement drift of the UAV can be compensated by a fixed point or high pass filter. Finally, two displacement drift compensation methods are used for comparison, and the time-domain comparison results are shown in Figure 9.

It can be seen from Figure 9 that the displacement drift generated by the UAV hovering can be removed by the background fixed point and high-pass filtering. Then, the high-pass filter value in Figure 9b is closer to the actual value, but the displacement recognition result for the amplitude is not good. Affected by the displacement increment of the fixed point, the amplitude is higher or lower in some times. Figure 10 shows an image of the processed displacement time-history map after a fast Fourier transform to the frequency domain.

From the comparison in the frequency domain from Figure 10, it can be seen that both the background point displacement compensation and the filtering denoising have good effects in the frequency domain, which are completely consistent with LDS. The two types of UAV hovering monitoring displacement compensation have their respective advantages and disadvantages. Although the displacement compensation based on the background fixed point has a better effect on detail recognition, it results in poor time-

domain monitoring, which is affected by the UAV trajectory caused by the different selection of angles and background points during UAV monitoring. The high-pass filtering time-domain recognition effect is better, but if the natural vibration frequency of the structure is included in the noise generated by the UAV lens, a recognition error will occur. The natural vibration frequency of the experimental model adopted in this paper is not within the range of the UAV lens noise, so high-pass filtering is used to process the drift caused by UAV hovering.

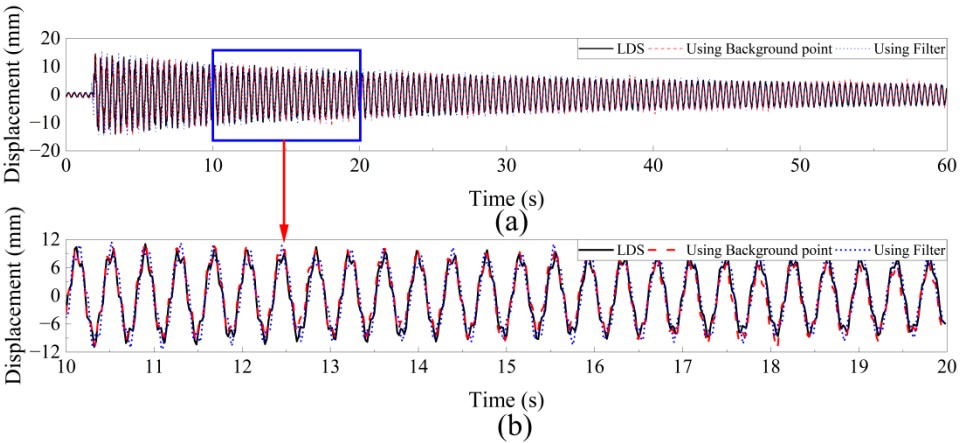

**Figure 9.** Time-domain comparison of UAV monitoring time history denoising processing: (**a**) full displacement monitoring; (**b**) 10~12 s displacement monitoring.

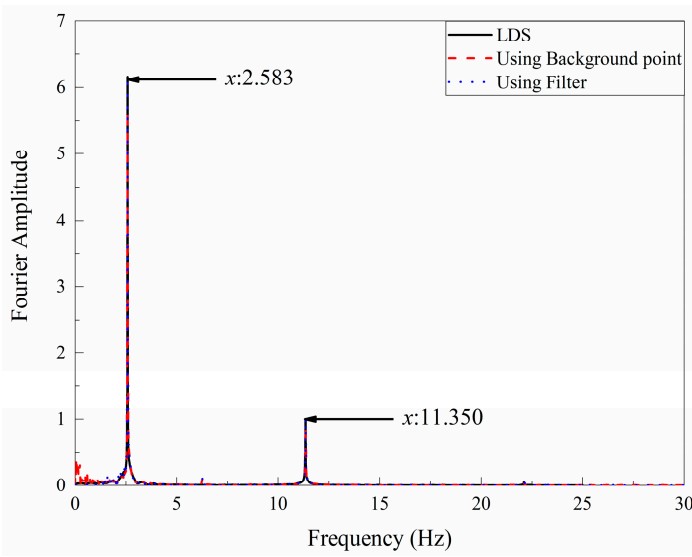

**Figure 10.** Frequency domain comparison of UAV monitoring time history with and without denoising process.

## 3. Monitoring Principle Based on Target-Free DSST Vision Algorithm

Images can remotely encode information within the field of view in a non-contact manner to obtain data for structural health monitoring, potentially addressing the problem in monitoring using contact sensors. The video encodes the individual images arranged in the time dimension, and then examines the image characteristic information for structural health monitoring.

The large-scale structural health monitoring of UAV equipped with vision algorithm can be divided into five steps: camera calibration, target tracking, UAV hover processing, adaptive scale factor acquisition and system identification, as shown in Figure 11. The previous section solved the problem of displacement drift caused by UAV hovering monitoring, which will not be repeated here.

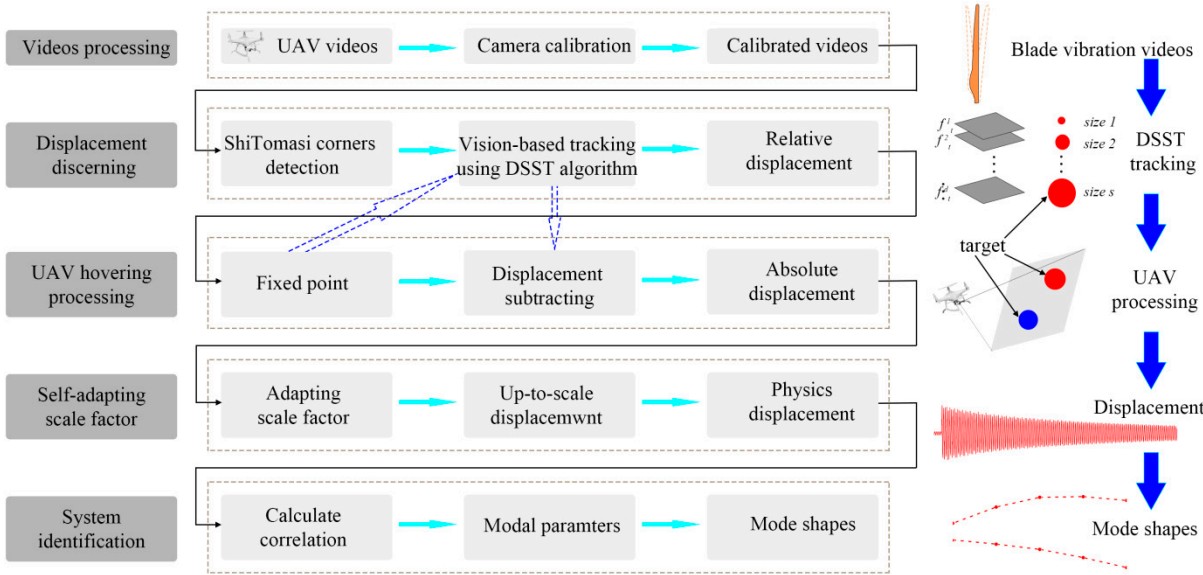

**Figure 11.** Target-free DSST vision algorithm and general steps for UAV monitoring of dynamic properties of structures.

### 3.1. Camera Calibration

Camera calibration is to shoot the calibration plate by the camera, determine the internal and external parameters of the camera with the intrinsic value of the characteristic points of the calibration plate, and then convert the image coordinates into physical coordinates through the scale factor. The general equation for converting from image coordinates to physical coordinates is as follows:

$$\begin{pmatrix} x \\ y \\ 1 \end{pmatrix} = \begin{bmatrix} f_x & \gamma & c_x \\ 0 & f_y & c_y \\ 0 & 0 & 1 \end{bmatrix} \begin{bmatrix} r_{11} & r_{12} & r_{13} & t_1 \\ r_{21} & r_{22} & r_{23} & t_2 \\ r_{31} & r_{32} & r_{33} & t_3 \end{bmatrix} \begin{bmatrix} X \\ Y \\ Z \\ 1 \end{bmatrix} \tag{5}$$

The simplified expression is:

$$sx = K[R|t]X \tag{6}$$

where $s$ is the scale factor; $(x, y, z, 1)^{\mathrm{T}}$ is the image coordinate; $K$ is the camera internal parameter representing the projection transformation from the three-dimensional real world to the two-dimensional image; $(X, Y, Z, 1)^{\mathrm{T}}$ is the world coordinates; in the intrinsic parameters, $f_x$ and $f_y$ are the focal lengths of the camera in the horizontal and vertical directions; $c_x$ and $c_y$ are the offsets of the optical axis in the horizontal and vertical directions; $\gamma$ is the tilt factor; $R$ and $t$ are the camera The external parameters represent rigid rotation and translation from 3D real-world coordinates to 3D camera coordinates; $r_{ij}$ and $t_i$ are elements of $R$ and $t$, respectively.

In this paper, the calibration method in the paper [26] is used to calibrate the UAV lens; the video is calibrated by the camera's internal parameters, the tangential and radial distortions. The calibration of image distortion through camera calibration can effectively eliminate image distortion and result in more accurate displacement measurement.

### 3.2. Target Tracking with Target-Free DSST Vision Algorithm

Vision-based structural health monitoring in the field of civil engineering mostly uses digital image correlation (DIC) technology [27], template matching [28], color matching [29], optical flow method [30] and other algorithms for target tracking. Restricted to the conditions of additional structural markers, constant illumination, and single background, under

the actual engineering background conditions, the above-mentioned limitations can be solved to accurately monitor the actual engineering structure.

In 2014, Danelljan et al. [31] improved the KCF algorithm and proposed a robust scale estimation method based on the detection and tracking framework, namely the DSST (Discriminative Scale Space Tracker) algorithm. The DSST algorithm can learn translation and scale estimation by separate filters and the target is tracked with a position filter. This algorithm improves the accuracy of the exhaustive scale space search method, and the running speed can reach 25 times the frame rate, which is very effective for fast structural health monitoring.

First, a discriminative correlation filter is trained to locate the target in a new frame, using target images in several grayscale images as training samples, marked as filters that require correlation output. The optimal correlation filter obtained by the minimized error emissions and the resulting time compensation satisfies the following equation:

$$\varepsilon = \sum_{j=1}^{t} \|h_t * f_j - g_j\|^2 = \frac{1}{MN} \sum_{j=1}^{t} \|\overline{H_t}F_j - G_j\|^2 \tag{7}$$

where the functions $f_j$, $g_j$ and $h_t$ are of size $M \times N$. $*$, which represents a cyclic correlation, and the second equal sign follows from Parseval's theorem. Capital letters denote discrete Fourier transforms (DFTs) of the corresponding functions, and the overline in $\overline{H_t}$ denotes the complex conjugate. The result of minimizing the above equation can be obtained:

$$H_t = \frac{\sum_{j}^{t} \overline{g_j} F_j}{\sum_{j=1}^{t} \overline{F_j} F_j} \tag{8}$$

In engineering practice, $\overline{H_t}$ is usually divided into numerator $A_t$ and denominator $B_t$ for iterative update operations.

For the image patch z of $M \times N$ in a new frame of image, its response score is $y$, and the calculation method is as follows:

$$y = \xi^{-1}\{\overline{H_t}Z\} \tag{9}$$

where $\xi^{-1}$ is the inverse DFT operator. The largest value in $y$ is considered as an estimate of the new position of the target.

Considering the multi-dimensional features of the image, let $f$ be the feature, $f$ has $d$ dimensions, $f^l$ is the $l$ dimension, and the value of 1 is 1 to $d$, then the minimized loss function is:

$$\varepsilon = \|\sum_{l=1}^{d} h^l * f^l - g\|^2 + \lambda \sum_{l=1}^{d} \|h^l\|^2 \tag{10}$$

where $\varepsilon$ is the regular term. In the above equation, only one training sample is considered, and $H$ can be obtained by solving:

$$H^l = \frac{\overline{G}F^l}{\sum_{k=1}^{d} \overline{F^k}F^k + \lambda} \tag{11}$$

Similarly, $H$ is split into numerator $A$ and denominator $B$ and updated iteratively as follows:

$$A^l{}_t = (1 - \eta)A^l{}_{t-1} + \eta \overline{G_t}F^l{}_t \tag{12}$$

$$B_t = (1 - \eta)B_{t-1} + \eta \sum_{k=1}^{d} \overline{F^k}{}_t F^k{}_t \tag{13}$$

where $\eta$ is the learning rate. For patch Z in the new image, its response score is $y$, and the calculation method is as follows:

$$B_t = (1 - \eta)B_{t-1} + \eta \sum_{k=1}^{d} \overline{F^k}_t F^k{}_t \tag{14}$$

$$y = \xi^{-1} \left\{ \frac{\sum_{l=1}^{d} \overline{A^l} Z^l}{B + \lambda} \right\} \tag{15}$$

where the maximum value of $y$ is considered to be the estimate of the target's new position.

The traditional DSST algorithm performs position tracking and scale estimation on the target by constructing a position filter and a scale filter, and only a target with strong features can be tracked. In this paper, the machine learning method is used to train the position filter and the scale filter to enhance the target image features, and a target-free DSST visual tracking algorithm is proposed. The target tracking steps based on the label-free DSST algorithm are shown in Figure 12.

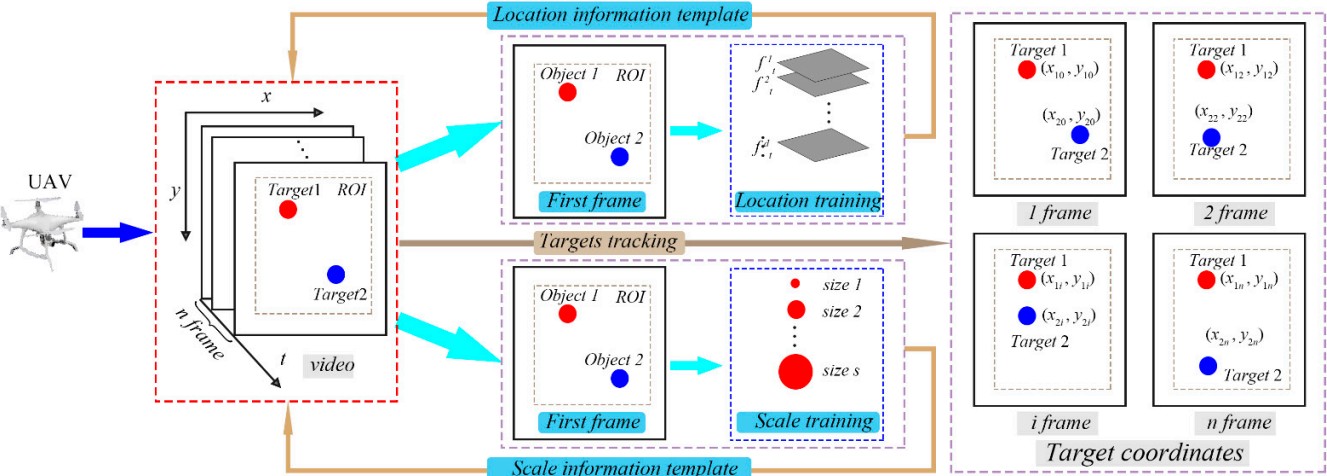

**Figure 12.** Target-free DSST vision algorithm tracking principle.

As described in the figure, tracking steps are as follows:

Step 1: Create a Region of Interest (ROI). To reduce the amount of calculation and eliminate unnecessary information in the image, an ROI area is established in the image.

Step 2: Perform scale training. After the target is selected, the scale change correlation operation is performed, and the scale information is obtained by S-scale training based on the selected target sample of the image.

Step 3: Perform the position training at the same time as scale training. Taking the selected image as the sample component of the d-layer feature pyramid, each feature sample is trained to obtain the maximum response output, and then the new target center position is obtained to determine the position information.

Step 4: A model update. The parameters obtained by scale training and position training update the position filter and scale-space filter of the original model to prepare for the next step of target tracking.

Step 5: Target Tracking. The target tracking is performed on each frame of an image with the parameters of the updated model. In each frame, the target with the largest response is found according to the correlation operation to obtain a new target position and scale, and finally the coordinates are output through the tracking frame.

In this paper, the selected target is used to train the scale and position of the frame for machine learning, and the obtained target features are strengthened by machine learning training to achieve the purpose of edge and corner tracking of the structure and even target-free monitoring can be achieved.

### 3.3. Relative Displacement Calculation

Since the UAV itself influences displacement drift when hovering, relative coordinates and displacement can be obtained by target tracking. In each frame of the time series, the corresponding coordinate $P_i(x_i, y_i)$ of the target and the target coordinate $P_0(x_0, y_0)$ of the first frame can be calculated by the following equation to find the relative displacement $d_R$ of the structure (unit: pixel):

$$d_R = P_i - P_0 \tag{16}$$

### 3.4. System Identification

Compared with the fixed camera, the advantages of UAV to shoot structural vibration videos go beyond finding a favorable position for monitoring and eliminating the image distortion caused by the atmosphere to the camera.The flexibility of UAV also permits a close approach to the structure, so as to maximize the resolution and vibration amplitude of UAV and achieve more accurate structural health monitoring. After the UAV displacement drift compensation is realized, the absolute displacement of the structure is calculated by Equation (4), and the displacement time history diagram of multiple monitoring points of the structure is obtained. Finally, the global modal mode shape of the structure is calculated by the response.

## 4. Experimental Verification Based on Target-Free DSST Vision Algorithm

At this stage, many vision algorithms have emerged for the dynamic characteristic test based on vision, but most algorithms need certain special conditions for the structure. To obtain fast, convenient and low cost results, this paper verifies the optical flow method [30], color matching method [29] and target-free DSST algorithm commonly used by researchers for visual health monitoring.

### 4.1. Test Equipment

In order to quantify the accuracy of the algorithm, this experiment uses a Canon R6 camera with a 24–105 mm zoom lens with its the video resolution to 720P/1080P/4K and frame rate 50 fps, simulating the scaled wind turbine model, light source, tripod, etc. in the shutdown state. In order to verify the accuracy of computer vision monitoring of structural displacement, a Banner250U laser displacement sensor with the sampling frequency of 50 Hz was used here. The test equipment is shown in Figure 13.

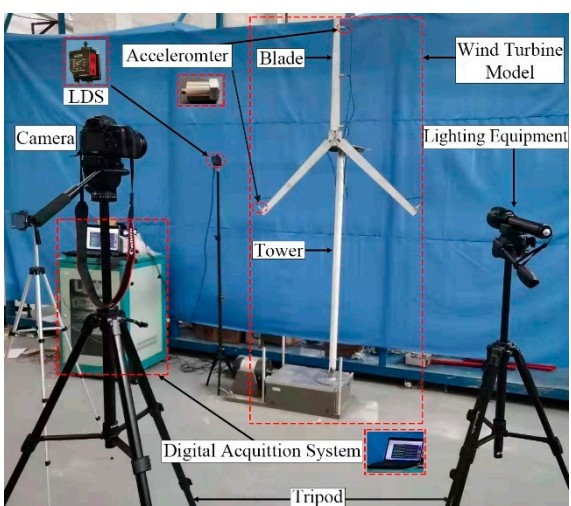

**Figure 13.** Test equipment.

### 4.2. Algorithm Testing under Simulated Engineering Conditions

The existing tracking algorithms tests are usually carried out under ideal conditions. Because of the harsh wind field environment and tall blades, artificial markers are difficult

to instal. As a result, conventional tracking algorithms cannot perform good tracking work. To verify the practicability of the target-free DSST algorithm in the actual wind farm environment, this experiment performs visual monitoring under simulated conditions including idealization, illumination change and complex background. The optical flow method and the color matching method have the obvious limitations of adding artificial markers as features during the monitoring process. However, the target-free DSST algorithm uses the inherent corners or edges of leaves for tracking.

The camera is used to monitor the states among three simulated machines during 90 s, in which the blue cloth is used as the background of the ideal environment, the lighting equipment is used to irradiate the blades from weak to strong in 16~17 s, and the internal environment of the laboratory is used as the engineering background for the identification of dynamic characteristics. In the test project with ideal conditions, the tracking effects of the three algorithms are good due to the simple background with no external interference. The monitoring results are shown in the next section, so they are not repeated here. The algorithm performance test focuses on the monitoring of illumination changes and complex background. The performance of the algorithm in the different simulated wind field environments is shown in Figures 14–16.

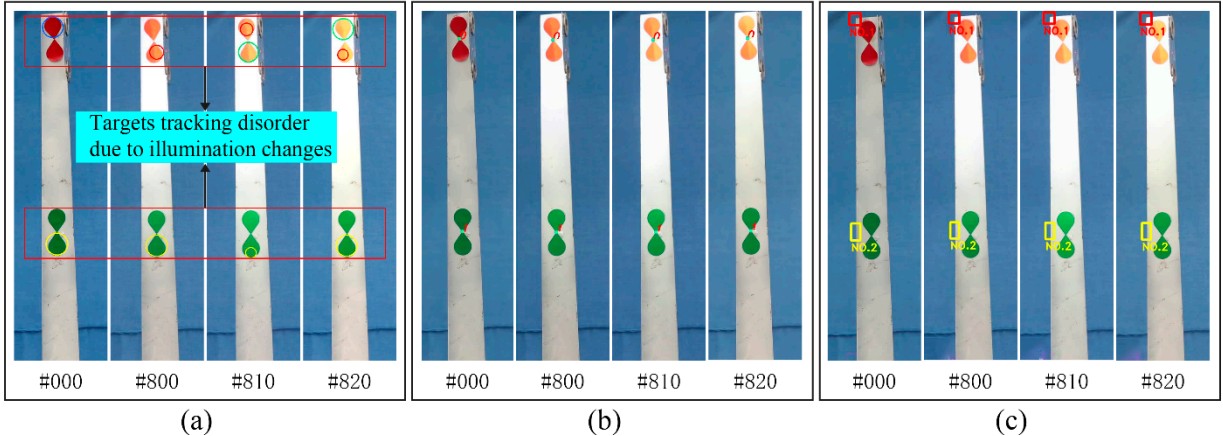

**Figure 14.** Different algorithms monitor the tracking situation of the algorithm under the condition of illumination change: (**a**) color matching algorithm; (**b**) optical flow method; (**c**) target-free DSST algorithm.

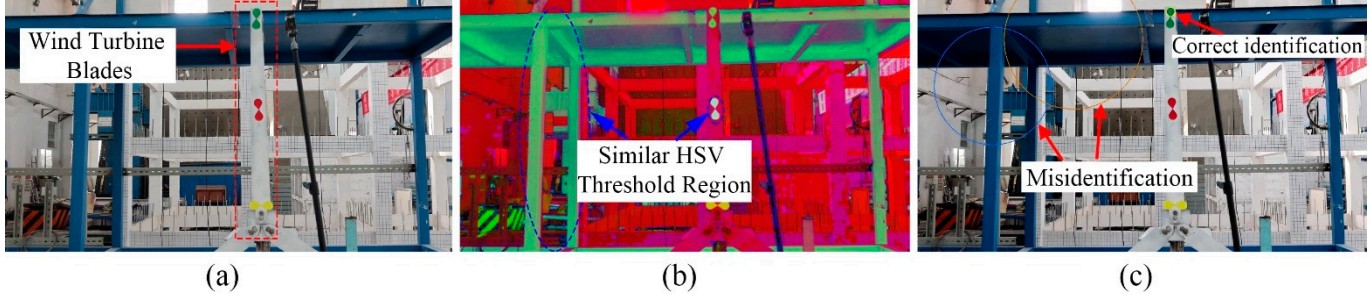

**Figure 15.** Monitoring effect of the color matching algorithm with a complex background: (**a**) wind turbine blade under complex background; (**b**) HSV color model image effect; (**c**) actual monitoring effect of the color matching algorithm.

Figure 14 shows the performance of the three algorithms in the tracking process under the condition of illumination change. When the illumination changes, the predetermined threshold of the monitoring points exceeds the range, and the color matching algorithm cannot capture the target points. Figure 14a shows the tracking situation of the color matching algorithm at different frame numbers. It can be seen that the color matching

algorithm cannot track the monitoring target as long as the illumination changes occur, such as the 800th, 810th and 820th frames. The color matching algorithm is particularly sensitive to illumination changing and cannot track the object when the illumination changes. The optical flow method and the target-free DSST algorithm are not sensitive to illumination when tracking, so they can both accurately capture the target point.

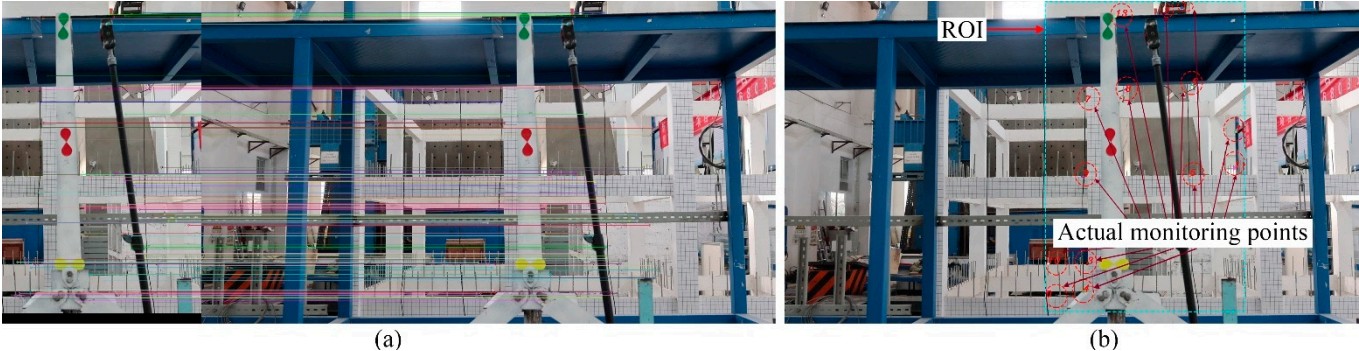

**Figure 16.** Monitoring effect of optical flow method in a complex background situation: (**a**) Harris corner matching in ROI area; (**b**) actual monitoring effect of optical flow method in ROI area.

Figures 15 and 16 show the monitoring performance of the color matching algorithm and optical flow method with complex background settings. It can be seen in Figure 15 that the former algorithm has serious error in recognition when transforming RGB color space into HSV space under the complex background condition. It can be tracked only when the color difference between the monitoring point and the background is large, which is not applicable in practical engineering. From Figure 16, it can be seen that most of the identified corners in the ROI selection under the complex background condition are background points, and 100 corners are set in the figure for matching. According to the matching results in Figure 16b, the strong Harris corners fall on the blades is less, which will lead to the inaccurate identification of the monitoring points including to the extent of some being identified as useless points. This results in large amounts of calculation and too much data redundancy in memory, and thus is not suitable for actual monitoring.

The target-free DSST algorithm performs well in different actual monitoring environments attributed to the scale and location training of machine learning, which does not require the artificial markers in Figure 14.

### 4.3. Analysis of Monitoring Results

The wind turbine model vibrates freely under the effect of excitation which the three algorithms are used to track. The three algorithms are used to compare LDS, the displacement time history curve and frequency domain information under different conditions as shown in Figure 17. Different equipment will produce time differences in monitoring. In this paper, the time point matching of the first peak measured in the time domain is used to solve the problem of displacement time history phase deviation caused by time difference [28].

It can be seen from Figure 17 that the three algorithms perform well in both the time and frequency domains under ideal conditions, which means these algorithms are available for health monitoring under ideal conditions. A light source is used to irradiate the tip of the wind turbine blade structure according to the change of illumination, and excitation is applied to the tip to make it vibrate freely. During the period of 16–17 s of artificial light, as shown in Figure 14, the displacement time history under the condition of light change in Figure 17 is interrupted three times, and there is a disorder in the frequency domain. The reason that the color matching algorithm and optical flow method under complex backgrounds are not suitable for monitoring is explained in Section 4.2. Therefore, only the target-free DSST algorithm works in tracking object structure under complex background conditions, the curve of which is still robust in Figure 17.

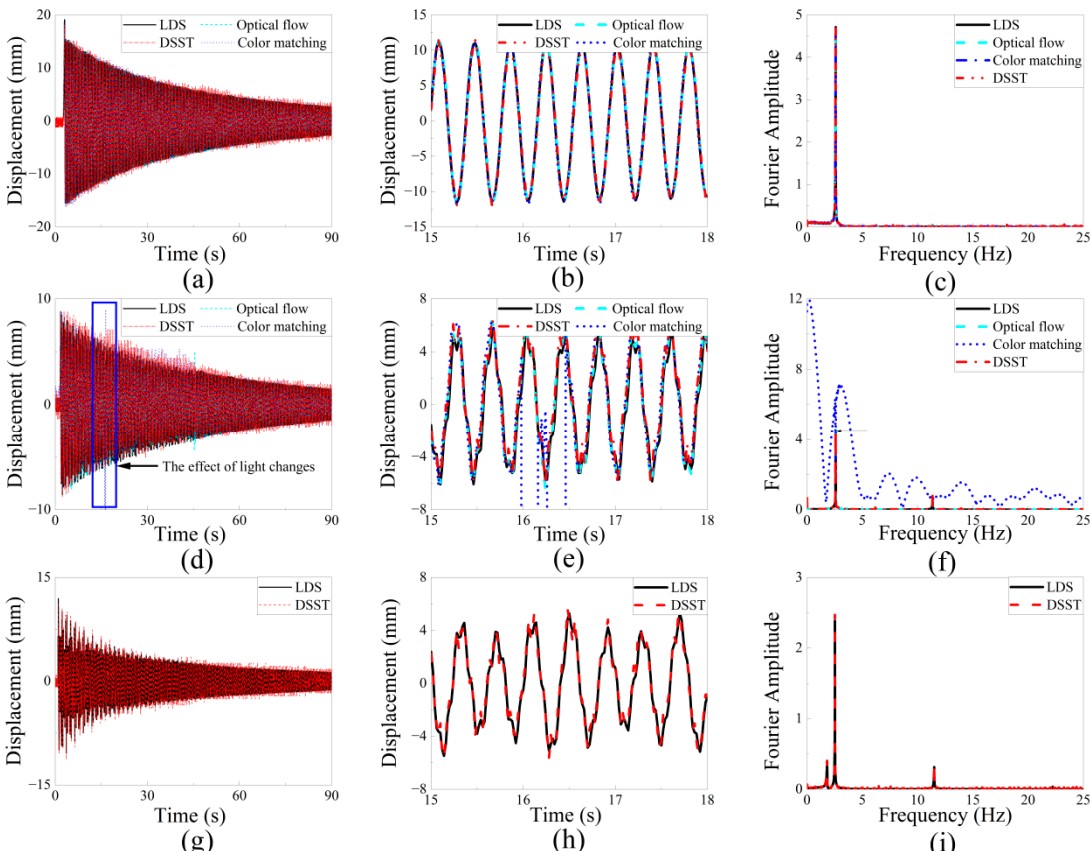

**Figure 17.** Comparison of monitoring results of three algorithms under different conditions: (**a**–**c**) ideal conditions; (**d**–**f**) light change conditions; (**g**–**i**) complex background conditions.

### 4.4. Monitoring Error Analysis

Taking the LDS data as the benchmark, the displacement data collected from different algorithms are used to perform error analysis [32,33]. The absolute errors and the error distributions of the three algorithms under different conditions are shown in Figure 18.

Figure 18 shows the absolute error and the PDF value calculated from the three algorithms under different conditions. It can be seen from the figure that the target-free DSST algorithm has relatively smaller monitoring errors compared with the others in the three simulated actual environments. Under varied lighting conditions, the color matching algorithm has a large error due to the monitoring disorder caused by lighting changes. Under complex background conditions, neither both the optical flow method nor the color matching algorithm can succeed in the tracking process. However, the target-free DSST algorithm achieves an excellent performance controlling the error within 1 mm, which meets the monitoring needs.

To quantify the accuracy of the algorithm, this paper introduces the root mean square error (RMSE), the correlation coefficient ($\rho$), and the coefficient of determination ($R^2$) for error analysis. The equations are:

$$\text{RMSE} = \sqrt{\sum_i \left(x_v(i) - x_s(i)\right)^2 / n} \tag{17}$$

$$\rho = \frac{\left|\sum_i \left(x_s(i) - \mu_s\right) \times \left(x_v(i) - \mu_v\right)\right|}{\sqrt{\sum_i \left(x_s(i) - \mu_s\right)^2} \sqrt{\sum_i \left(x_v(i) - \mu_v\right)^2}} \tag{18}$$

$$R^2 = 1 - \frac{\sum_i \left(x_v(i) - x_s(i)\right)^2}{\sum_i \left(x_s(i) - \mu_s\right)^2} \tag{19}$$

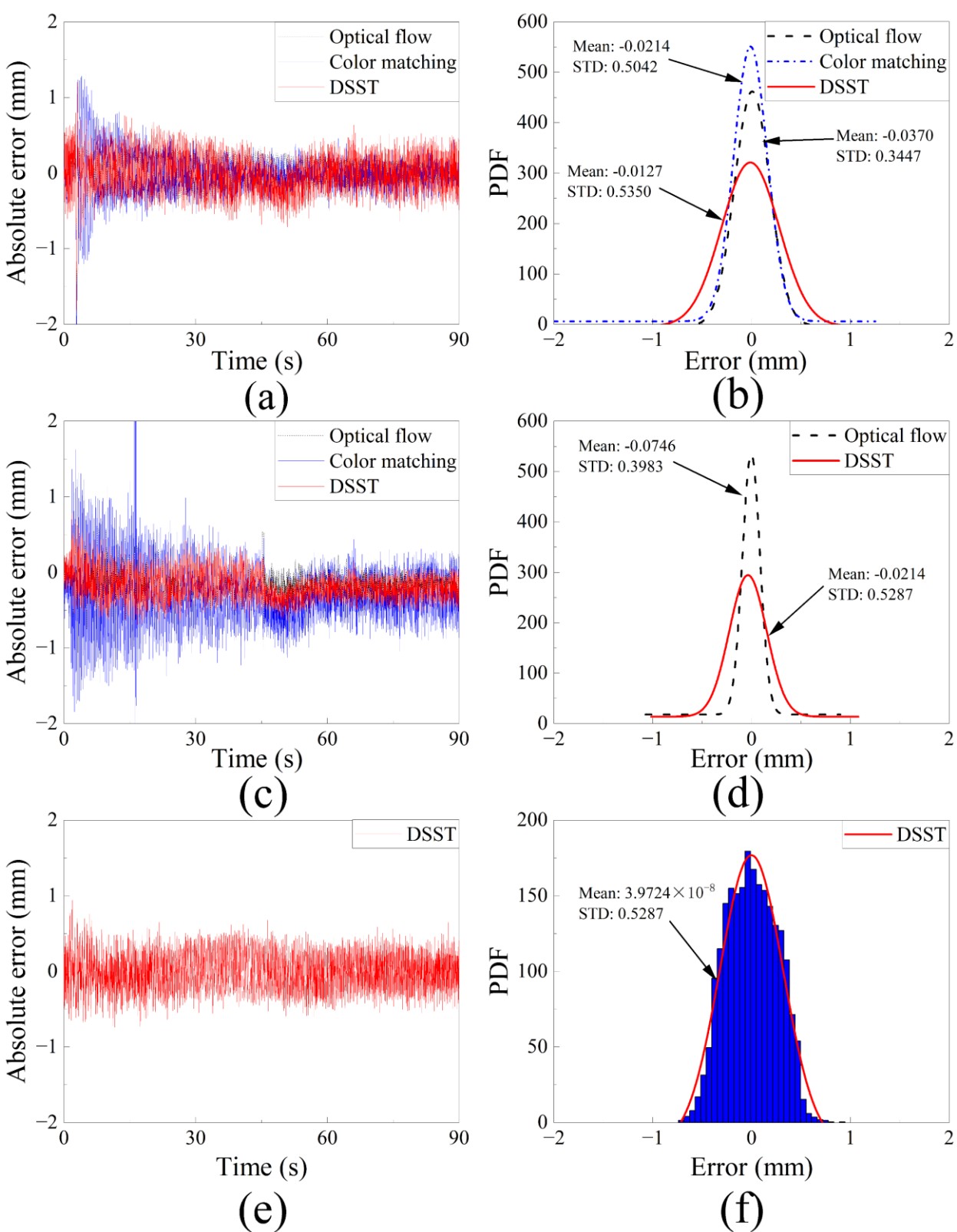

**Figure 18.** The absolute errors of three algorithms are compared with PDF under different conditions: (**a**,**b**) ideal conditions; (**c**,**d**) illumination change conditions; (**e**,**f**) complex background conditions.

RMSE is calculated using Equation (17), where $n$ is the total number of monitoring, $x_v$ and $x_s$ are displacement data from vision monitoring and laser displacement sensors, respectively, which measure the deviation between the measured value and the reference

value. $\rho$ is calculated by Formula (18), where $\mu_v$ and $\mu_s$ are the average values of the two displacement trajectories, $\rho$ varies from 0 to 1 unit, $\rho = 1$ represents complete correlation, and $\rho = 0$ means that the two recording tracks have no correlation. The calculation formula of $R^2$ is Formula (19), which is used to determine the matching degree of the two recorded tracks. $R^2$ also belongs to [0, 1], and the unit represents the similarity of the two recorded tracks [32]. The comparison of vibration monitoring data errors of the three algorithms under different conditions is shown in Table 1.

**Table 1.** Displacement time history and frequency domain error of different algorithms under different engineering conditions.

| Algorithms Conditions | Optical Flow Method | | | Color Matching Algorithm | | | Target-Free DSST Algorithm | | |
|---|---|---|---|---|---|---|---|---|---|
| | RMSE | $\rho$ | $R^2$ | RMSE | $\rho$ | $R^2$ | RMSE | $\rho$ | $R^2$ |
| ideal condition | 0.3447 | 0.9980 | 0.9959 | 0.5046 | 0.9960 | 0.9913 | 0.5351 | 0.9951 | 0.9902 |
| illumination variation | 0.4052 | 0.9892 | 0.9766 | 50.4127 | 0.0031 | 0.00015 | 0.4669 | 0.9807 | 0.9782 |
| complex background | - | - | - | - | - | - | 0.4142 | 0.9861 | 0.9813 |

It can be seen from Table 1 that under ideal conditions, the three algorithms have high monitoring accuracy. Under the condition of illumination change, the optical flow method and the target-free DSST algorithm have high accuracy, but the color matching algorithm has large errors and poor correlation, meaning that the color matching algorithm cannot monitor under changing lighting conditions. Both the optical flow method and the color matching algorithm cannot monitor under complex background conditions. The target-free DSST algorithm has high monitoring accuracy and good correlation due to its good robustness. Overall, the target-free DSST algorithm can meet the monitoring task request under different conditions, with the accuracy meeting the engineering requirements.

Vibration monitoring experiments of optical flow method, color matching algorithm and target-free DSST algorithm were carried out in different simulation environments, and the robustness of the target-free DSST algorithm was verified. Compared with other algorithms, the target-free DSST algorithm has the following advantages: (1) It is suitable for monitoring the actual engineering environment (including illumination changes and complex backgrounds). (2) Without additional artificial targets, it can be accurately monitored by using the structure's corners or edges features, which expands the application range of vision-based monitoring of large-scale structures, especially large-scale wind turbines. (3) The monitoring error can meet the needs of structural health monitoring. The monitoring results of different actual projects show that the absolute error is within 1 mm, the root-mean-square error is below 0.5, and the correlation can be above 0.97, which meets the requirements of engineering health monitoring. Given the good robustness of the target-free DSST algorithm, it was selected as the visual monitoring algorithm for monitoring in subsequent experiments.

## 5. Wind Turbine Blade Structure Vibration Test

In order to verify the monitoring effect of the above-mentioned visual algorithm combined with the UAV on the wind turbine, the structural vibration test of the wind turbine blade is now carried out. There are various vibration forms of wind turbine blades under the effect of external loads, the main vibration modes of which are edgewise, flap-wise and torsion, as shown in Figure 19. However, due to the small torsional deformation of wind turbine blades, it is generally not considered. Thus, only the edgewise and flap-wise mode of vibration are considered for wind turbine blades herein. Given that the wind turbine's sway and swing vibrations are very small compared to the overall size of the wind turbine, this paper only considers the vibration in the X-direction.

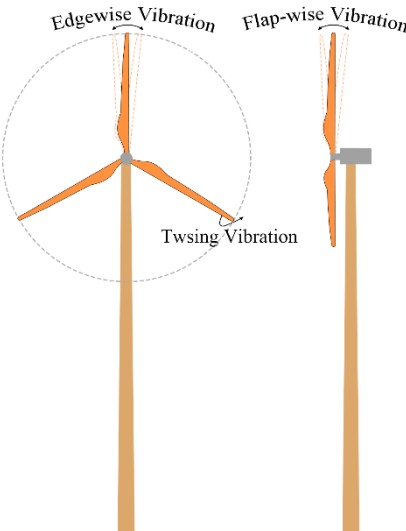

**Figure 19.** Basic vibration forms of wind turbine blades.

### 5.1. Experimental Equipment

A DJI Phantom 4Pro UAV with a lens resolution of 720P/1080P/4K and a frame rate of 30/60/120 fps is used in this test. To verify the reliability of the UAV, a Canon R6 camera with a 24~105 mm zoom lens and mobile phone are also chosen in the test to compare and verify, in which the lens distortion is corrected by the method described in Section 3.1 for shooting videos. The vibration test is carried out on the scaled wind turbine model under simulated shutdown, as shown in Figure 20. To verify the accuracy of the visual test, the LDS and INV9812 accelerometers of Banner250U are used with the sampling frequency to 50 Hz, and the traditional vibration test digital system composed of their corresponding equipment is used for further verification.

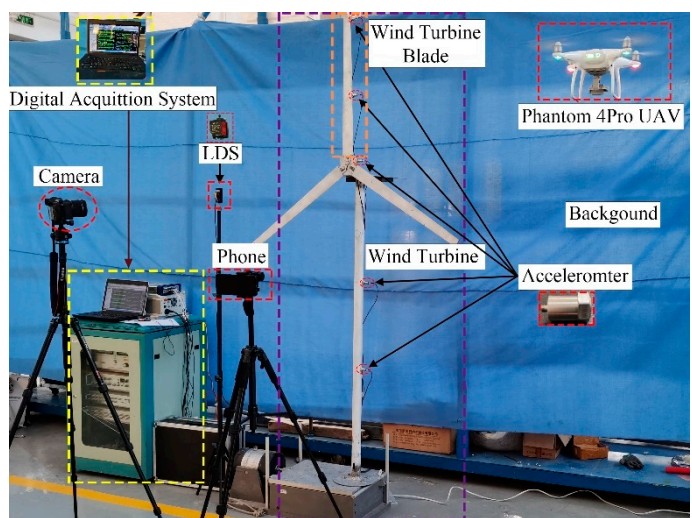

**Figure 20.** Experimental equipment.

### 5.2. Edgewise and Flap-Wise Direction Vibration Test

The scaled wind turbine model is applied with initial displacement in the edgewise and flap-wise directions, and then released to make it vibrate freely. The UAV hovers and monitors at 1.5 m to 2 m in the vertical direction of the wind turbine blade, with the frame rate of 60 fps and the resolution of 1080p. At the same time, the camera and mobile phone are used for vision comparison verification, with the frame rates of 50 fps and 60 fps, respectively. In order to verify the accuracy of visual monitoring, LDS is also used

for verification. The frequency is set to 50 Hz, and the displacement time history during 60 s are transformed into frequency domain information, as shown in Figure 21.

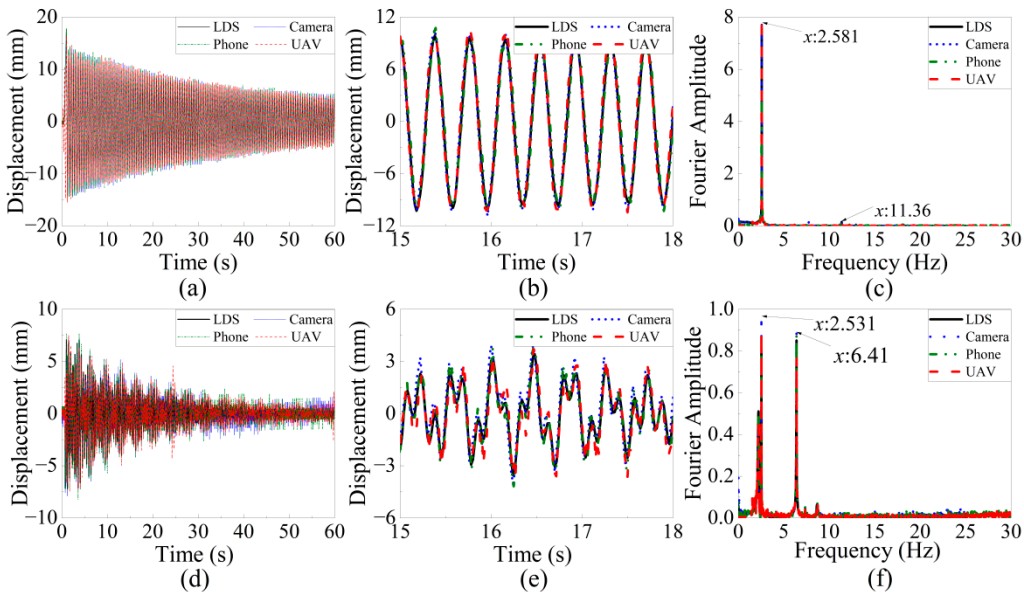

**Figure 21.** Wind turbine vibration time domain and frequency domain monitoring comparison: (**a**–**c**) edgewise direction; (**d**–**f**) flap-wise direction.

According to the monitoring information in Figure 21, generally speaking, the curves in the time and frequency domain achieved from the four devices are close to each other, and the phase is also consistent, so the monitoring effect is good. According to the frequency domain information shown in Figure 21c,f, the monitoring effects of the four devices are very close, and the first and second-order natural vibration frequencies are consistent, which proves the effectiveness of vision monitoring.

The natural vibration frequencies of the wind turbine in two vibration directions and UAV monitoring error with the LDS as the reference are shown in Table 2.

**Table 2.** Identification results of edgewise and flap-wise natural frequencies of wind turbine blades.

| Measurement Direction | Natural Frequency (Hz) | LDS | Phone | Camera | UAV | Error (UAV) |
|---|---|---|---|---|---|---|
| edgewise | 1th | 2.581 | 2.581 | 2.581 | 2.581 | 0% |
|  | 2th | 11.360 | 11.360 | 11.360 | 11.360 | 0% |
| flap-wise | 1th | 2.531 | 2.531 | 2.531 | 2.531 | 0% |
|  | 2th | 6.410 | 6.410 | 6.400 | 6.400 | 0% |

It can be seen from Table 2 that using the visual monitoring of the UAV and the target-free DSST algorithm, the first two natural vibration frequencies of the wind turbine in edgewise and flap-wise directions are consistent with those of the traditional LDS measurement with high accuracy, which can meet the actual engineering needs.

*5.3. Full-Field Monitoring and Accelerometer Verification*

The above tests verify the feasibility of computer vision monitoring of wind turbine blades, but in the actual monitoring, the response of multiple monitoring points of the structure is required. In addition to difficulties such as wiring, the cost of monitoring one point by one sensor is high. Therefore, realizing visual multi-point simultaneous monitoring can effectively reduce the monitoring cost. Vision measurement is now carried out for multiple monitoring points of the blade in this section. One LDS and three accelerometers are applied for verification, among which the LDS verifies point P1, the accelerometers

verify points P1, P3 and P5, and all five visual monitoring points monitor the swing directions. The test layout is shown in Figure 22.

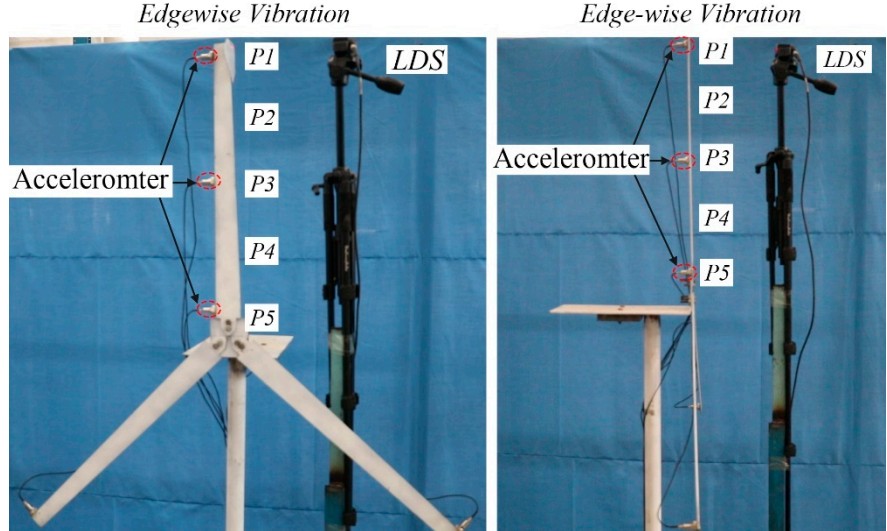

**Figure 22.** Schematic diagram of multi-point visual verification device.

Give the blade excitation at P1 to make it vibrate freely, the five points shown in Figure 22 are monitored visually, and the displacement time history monitored by LDs and vision is shown in Figure 23. The displacement time history peak values obtained by visual monitoring at the five monitoring points are shown in Table 3.

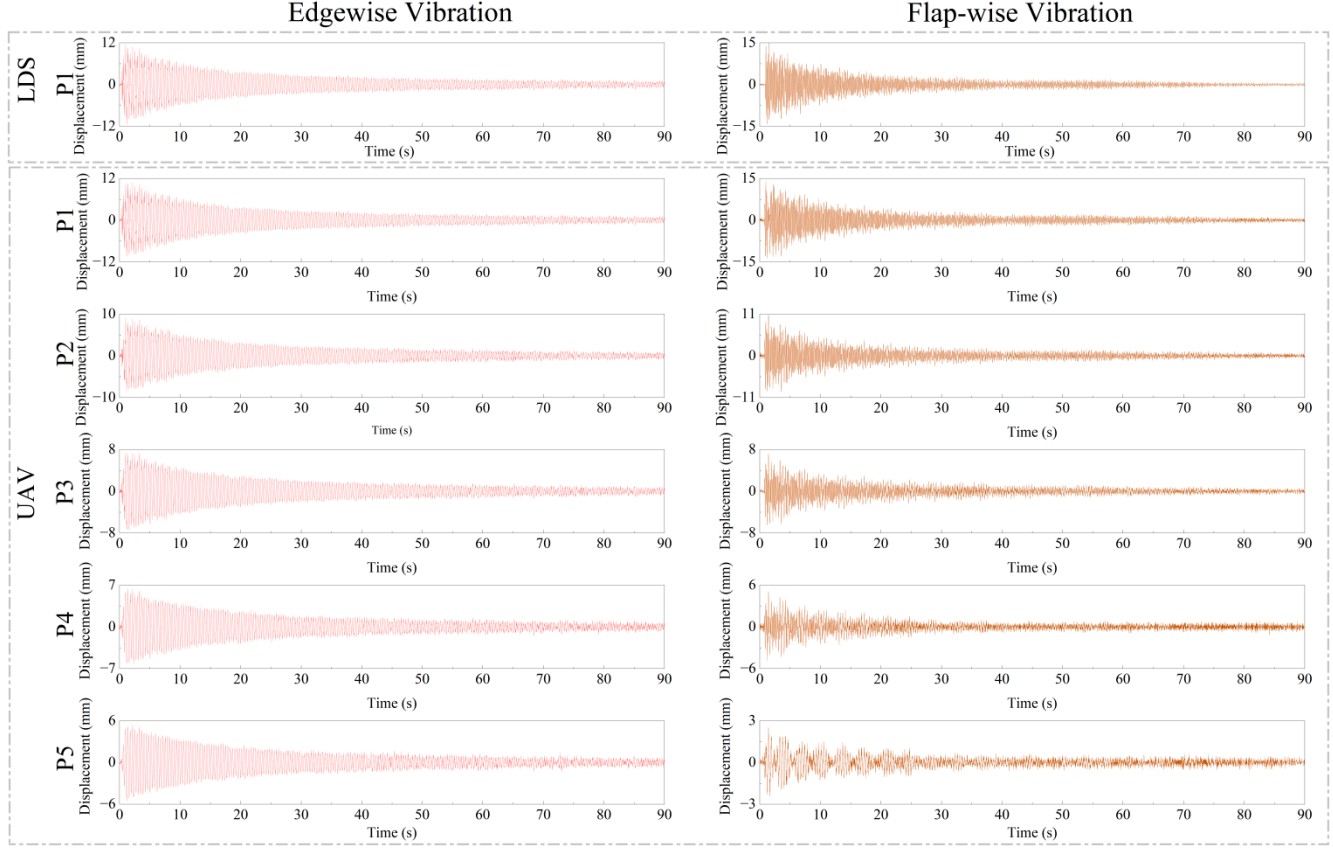

**Figure 23.** Comparison between vision multipoint monitoring and LDS displacement time history.

**Table 3.** Blade displacement peak value monitored in the whole field.

| Measuring Points | Edgewise (mm) | Flap-Wise (mm) |
|---|---|---|
| P1 | 10.859 | 13.821 |
| P2 | 9.005 | 10.736 |
| P3 | 7.394 | 7.156 |
| P4 | 6.335 | 5.046 |
| P5 | 5.366 | 2.490 |

It can be seen from Figure 23 that the displacement time-history trends of the five monitoring points in the edgewise and flap-wise directions are consistent, and the vision monitoring at point P1 is consistent with the LDS monitoring data, which verifies the feasibility of visual multi-point monitoring. It can be seen from Table 3 that the displacement peaks of the five monitoring points in the edgewise and flap-wise directions decrease sequentially from top to bottom, which is in line with the characteristics of structural vibration and confirms the accuracy of vision multi-point monitoring.

Accelerometers are favored by engineers because of their high sensitivity and wide frequency-domain measurement range, and have become a device commonly used in structural health monitoring. However, since the accelerometers needs to be attached to the surface of the structure, additional devices would need to be attached to the structure as well when installing the accelerometers. This approach will not only change the dynamic characteristics of the structure, but also cause damage to the structure. For large structures such as wind turbine blades, a more convenient, maintenance-friendly and non-destructive health monitoring method should be developed. Aiming at the inconvenience of accelerometers, this test uses vision and UAV to monitor the structure. At the same time, acceleration sensors and LDS are used to compare and verify visual accuracy.

For the rapid excitation of the fan model's edgewise and flap-wise directions, the UAV lens adopts 60 fps, and the accelerometers and LDS both use 50 Hz sampling frequency to make the comparison in the frequency domain. To better obtain the peaks at higher frequencies, the horizontal axis adopts a logarithmic scale, and the measured power spectral densities of three different devices in the two directions of edgewise and flap-wise are shown in Figure 24.

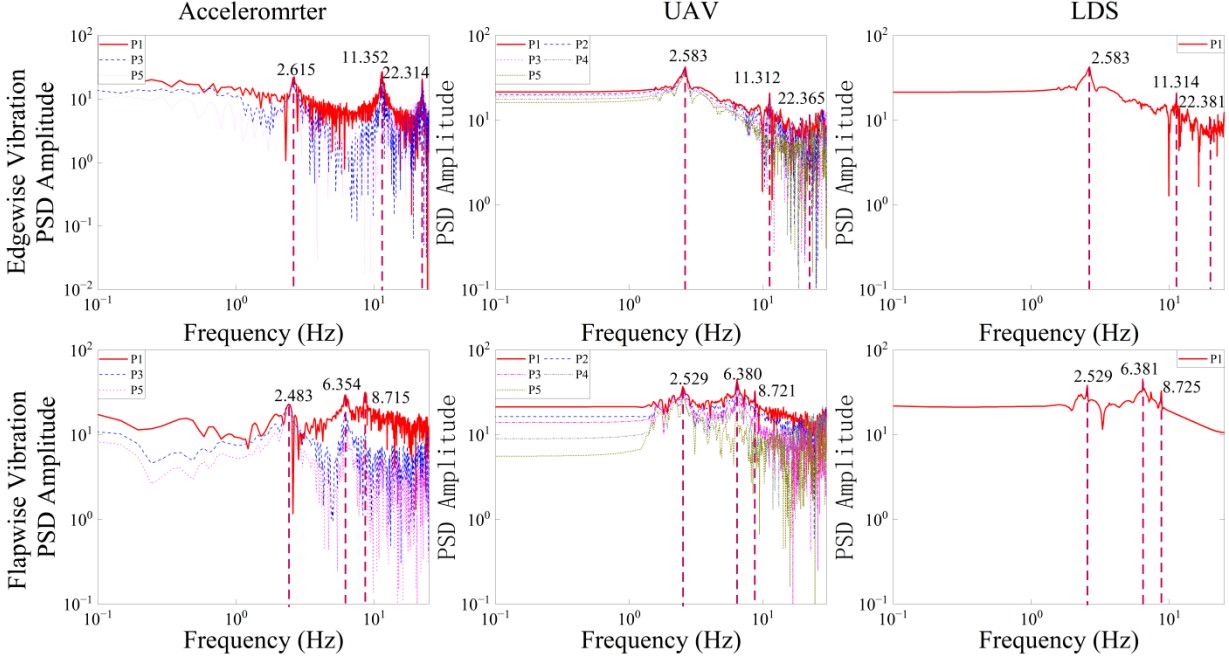

**Figure 24.** Frequency test results in edgewise and flap-wise direction.

It can be seen from Figure 24 that in the two directions of flap-wise and edgewise, the 1st and 2st natural frequencies can be clearly identified by vision and LDS. Especially in the flap-wise direction, the 3st natural frequency is better recognized by the accelerometer vision and LDS is difficult to identify. This result is reasonable because accelerometers are inherently sensitive to higher-order vibrations, whilst displacement sensors are more sensitive to lower-order vibrations. The higher-order displacement components are weakly excited under the free vibration test, while a higher frame rate lens performs well in capturing of small vibrations to identify higher-order frequencies.

Taking the date from the accelerometer as the benchmark, comparison in accuracy between the vision sensor and LDS is carried out. The natural frequency and error of edgewise and flap-wise are shown in Table 4. According to the calculation [34], the mode shape results are shown in Figure 25.

**Table 4.** Comparison of frequency domain recognition effects of different equipment.

| Measuring Equipment | Edgewise Natural Frequency (Hz) | | | Flap-Wise Natural Frequency (Hz) | | |
| --- | --- | --- | --- | --- | --- | --- |
| | 1st | 2st | 3rd | 1st | 2st | 3rd |
| accelerometer | 2.615 | 11.352 | 22.314 | 2.483 | 6.354 | 8.715 |
| LDS | 2.583 | 11.314 | 22.381 | 2.529 | 6.381 | 8.725 |
| UAV | 2.583 | 11.312 | 22.365 | 2.529 | 6.380 | 8.721 |
| error(UAV) | 1.22% | 0.35% | 0.23% | 1.85% | 0.41% | 0.07% |

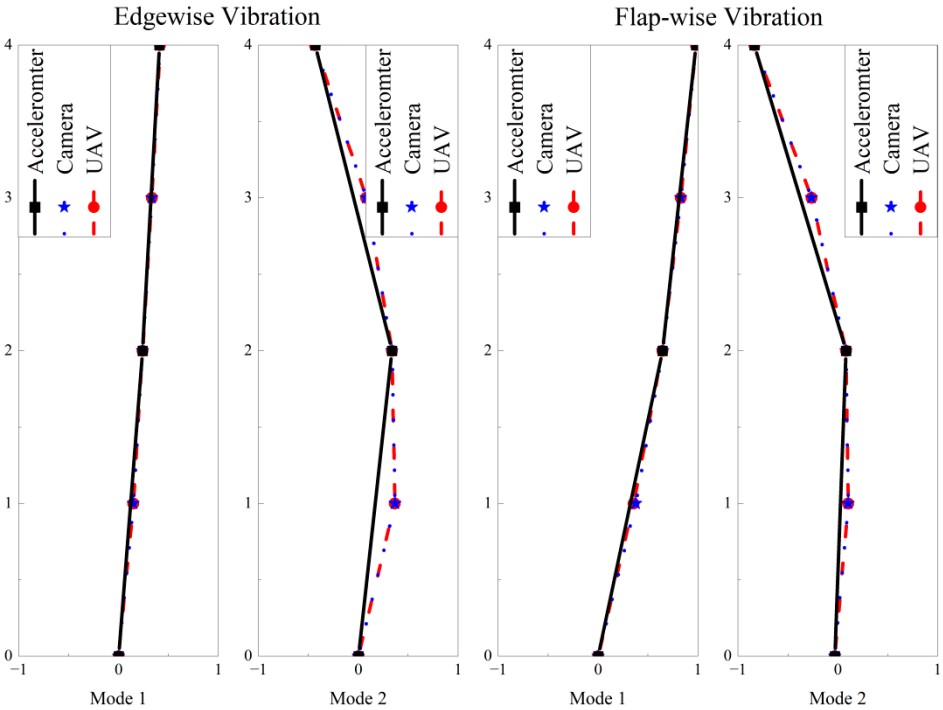

**Figure 25.** Edgewise and flap-wise direction vibration mode.

It can be seen from Table 4 that based on the benchmark data, the error of the measuring equipment reduces with increasing order of frequency regardless of edgewise or flap-wise mode of vibration, which is in line with the principle that the acceleration sensor is more sensitive to high frequency. Compared with the acceleration sensor, the maximum error of the measured natural vibration frequency in Table 4 is no more than 2%, and the test accuracy of the natural vibration frequency can reach more than 98%, which can meet the engineering needs. Therefore, the UAV vision sensor can be used to replace the accelerometers for large-scale structural health monitoring.

## 6. Conclusions

Focusing on the dynamic characteristics (i.e., natural frequencies) of large wind turbine blades, this study proposes a target-free vision algorithm, as well as a displacement compensation method for UAV hovering. First, the displacement drift caused by UAV hovering is investigated; in addition, the in-plane and out-of-plane displacements caused by UAV hovering monitoring are compensated by a high pass filter. Then, machine learning is employed to map the position and scale filters of the DSST algorithm to highlight the features of the target image. Subsequently, a target-free DSST vision algorithm is proposed; this algorithm is employed to extract the dynamic characteristics of the wind turbine blades. The following conclusions can be drawn from this study:

(1) Assuming the relative position of the rigid body remains unchanged, the adaptive scale factor can eliminate the Z-direction displacement drift out of the plane and convert the image coordinates into physical coordinates. Thus, the high-pass filter can effectively reduce the displacement drift generated in the X-Y plane during the UAV hovering.

(2) The proposed target-free DSST computer vision algorithm works well even with illumination changes, and a complex background. In addition, the target-free DSST visual algorithm can simultaneously monitor the spatial displacement of the whole blade. Accordingly, a smooth mode shape of the blade can be obtained.

(3) The proposed target-free DSST computer vision algorithm performs well in cooperation with the UAV. The vibration of the blade can be provided with high precision, both in time and frequency domains.

**Author Contributions:** Conceptualization, W.L. and W.Z.; methodology, W.L.; software, J.G. and W.Z.; validation, W.Z., W.L. and B.F.; formal analysis, W.Z.; investigation, W.L.; resources, W.L. and Y.D.; data curation, W.Z.; writing—original draft preparation, W.Z.; writing—review and editing, Y.D.; visualization, J.G.; supervision, W.L.; project administration, W.L.; funding acquisition, W.L. All authors have read and agreed to the published version of the manuscript.

**Funding:** This research was jointly funded by the National Natural Science Foundation of China (Nos. 52068049, 51908266), the Science Fund for Distinguished Young Scholars of Gansu Province (No. 21JR7RA267), and Hongliu Outstanding Young Talents Program of Lanzhou University of Technology.

**Data Availability Statement:** The datasets that support the findings of this study are available from the author (W.L.) upon a reasonable request.

**Conflicts of Interest:** The authors declare that they have no known competing financial interests or personal relationships that could have appeared to influence the work reported in this paper.

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
