# Peer review of "Dynamic Characteristics Monitoring of Large Wind Turbine Blades Based on Target-Free DSST Vision Algorithm and UAV"

_remotesensing, doi:10.3390/rs14133113_

Round 1
Reviewer 1 Report
The paper shows a very interesting research argument. The introduction is well addessed and the method is well explained. More accurate drawings and images could benefit the text understanding. Original results correspond with the research aims.
Reviewer 2 Report
Excellent work!
However, a minor language proof-reading is advised.
Reviewer 3 Report
The authors of the text deal with the problem of the flow of the liquid-air along the propeller blades. Traditional wind turbine blades use mainly manual inspection methods to evaluate the structural condition of the blades. These methods are time-consuming, laborious, costly and dangerous, in addition, the detection results are largely dependent on the technical expertise of the inspectors.
Due to the disadvantages of traditional wind turbine blade detection, a monitoring method based on Discriminative Scale Space Tracker (DSST) vision algorithm and the use of unmanned aerial vehicle (UAV) is proposed in this paper to test the dynamic characteristics of large wind turbine blades.
The UAV displacement rule in tracking, called "hanging in space", is evaluated and a displacement compensation method based on high-pass filtering and adaptive scaling factor is proposed to address the effect of space displacement in UAV hanging tracking.
Second, a machine learning method was proposed to be used to train the position and scale filters of the DSST algorithm to improve the image features of the target. The DSST vision algorithm has been tested in an engineering environment that simulates illumination changes and complex backgrounds, and the robustness of the algorithm is verified by comparative analysis with traditional computer vision algorithms.
Finally, the dynamic characteristics of wind turbine blades under simulated downtime were tested in combination with a UAV and the targetless DSST algorithm using indoor experiments. This verified the feasibility of the monitoring method proposed and described in the text in identifying the dynamic characteristics.
The results show that the proposed method can accurately identify the dynamic characteristics of the wind turbine blade structure and can realize a low-cost, non-contact and non-destructive method for monitoring the structural condition.
The paper is compiled based on scientific procedures, and is supplemented with graphical and mathematical apparatus.
From the formal point of view, I would recommend to adjust the font size of the description of axes, for example, Fig.4, 7, 9, 12, 13, 15,...18-24.
Author Response
Please see the attachment.

This manuscript is a resubmission of an earlier submission. The following is a list of the peer review reports and author responses from that submission.